

# The impact of scabies in tent cities in Kahramanmaraş after the Turkish earthquakes: oral pharmacologic treatment efficacy

Muhammed Mustafa Beyoğlu[1] and Mehmet Enes Gokler[2]

[1] Family Medicine, Kahramanmaraş Provincial Health Directorate, Kahramanmaraş, Turkey
[2] Department of Public Health, Ankara Yıldırım Beyazıt University Faculty of Medicine, Ankara, Turkey

Corresponding author
Muhammed Mustafa Beyoğlu, mstf-beyoglu@gmail.com

## ABSTRACT

**Background.** Our study was conducted to determine the impact of scabies in people living in collective living areas such as tent cities and container cities after the February 6 Kahramanmaraş earthquakes and to show the effectiveness of oral ivermectin treatment on scabies cases because topical treatments could not be used in this period when access to water was limited.

**Methods.** Among 233 patients diagnosed and treated with scabies in tent and container cities, 192 patients who met the criteria were included in the study. Descriptive statistics were given as number (n), percentage (%), mean, median, standard deviation (SD). In the comparison of categorical data, the chi-square test was applied; in the comparison of numerical data ANOVA analyses was applied.

**Results.** A total of 192 scabies patients (82.4%), 47.9% (N:92) of whom were women, were included in the study. The frequency of scabies in the total population in tent cities and container cities was found to be at least 0.54%. The most common symptoms were pruritis (99.0%) and rash (97.9%). The most common sites of lesion involvement were the umbilicus (87.0.%), forearm (75.0%), and back (70.3%). After the first dose of ivermectin, 159 (82.8%) patients showed complete recovery, while 30 (15.6%) patients showed partial recovery. 3 (1.6%) patients showed no improvement. After the second dose of ivermectin, 173 (90.1%) patients showed complete recovery. There were two (1.0%) patients who did not show improvement after two doses. Due to the deterioration of urban infrastructure after devastating earthquakes, the irregularity of mass living areas, the lack of hygiene conditions, and the difficulty of accessing clean, usable water, oral ivermectin may be the first choice for treatment in terms of ease of use and effectiveness.

## INTRODUCTION

Earthquakes, one of the most active geological processes on Earth, are also one of the most unpredictable natural disasters, with the potential to wreak destructive effects on populations and structures. According to the Center for Research in Disaster Epidemiology

(CRED) report, floods ranked first with 3,254 events, accounting for 44% of all disasters; storms ranked second with 2,043 events, accounting for 28% of all disasters; and devastating earthquakes ranked third with 552 events, accounting for 8% of all disasters (*Center for Research on the Epidemiology of Disasters, CRED)(2020*; *Mavrouli et al., 2023*). Despite representing only a small percentage of natural disasters, earthquakes are potentially fatal events, that can result in tens of thousands of deaths, or in individuals being buried under rubble or being made homeless (*Center for Research on the Epidemiology of Disasters, CRED)(2020*; *Clements & Casani, 2016*).

In the wake of devastating earthquakes, residents of the area affected are transferred by the authorities to communal accommodation areas to meet their needs for shelter (*Al Mandhari, 2023*). However, it is difficult to meet basic needs such as heating, water, toilets, and showers in communal areas such as tent and container cities in the immediate wake of such disasters due to damage to the urban infrastructure (*Villasana, 2023*). Another public health problem in communal living spaces is the emergence of acute dermatological symptoms and diseases due to poor hygiene conditions, physical injuries, increased skin contact with harmful agents, malnutrition and water scarcity (*Faria et al., 2008*). The impact of dermatological diseases associated with environmental and individual factors will inevitably increase during such periods (*Faria et al., 2008*; *Leung, Lam & Leong, 2020*). Scabies is an important dermatological disease of significant public health impact during these challenging times (*Engelman & Steer, 2018*; *Salavastru et al., 2017*).

It is a common skin disease affecting 150–200 million people each year (*Bernigaud, Fischer & Chosidow, 2020*). It is particularly prevalent in poor societies living in crowded environments, such as in Asia, Oceania, and Latin America, but in recent years its impact has also been increasing in Europe, following increased migration from regions with higher frequency, the rise of communal living spaces,the influx of refugees, and resistance to topical therapies (*Bernigaud, Fischer & Chosidow, 2020*). In Germany, a highly developed European country, the number of resistant cases treated as inpatients for scabies increased from 960 in 2012 to 10,072 in 2019 (*Sunderkötter, Wohlrab & Hamm, 2021*). According to another study, the incidence of scabies has increased by more than 300% in the last decade in Netherlands (*Van Deursen et al., 2022*). In another study, the increase in the number of collective living areas due to the frequency of earthquakes was shown as one of the reasons for this increase (*Engelman & Steer, 2018*). Although there are studies on the frequency of scabies being more common in collective living areas, there are no or very limited studies on increasing scabies cases due to collective living areas created after disasters.

Although we do not have data on the impact of scabies in the pre-earthquake period, factors such as adverse living conditions and poor hygiene in mass shelters such as tents and container cities established after the earthquakes measuring 7.7 and 7.6 Richter scales on February 6, 2023, in Kahramanmaraş and surrounding provinces in Turkey may have facilitated the emergence of scabies in the following period (*Tunalı, Harman & Özbilgin, 2023*).

In the literature, standard treatment protocols include permethrin, malathion, benzyl benzoate, sulfur ointment, and oral ivermectin (*Behera et al., 2021*). However, discomfort caused by topical treatment, inadequate application to local lesions, and poor compliance
with treatment among contacts of scabies cases are the biggest obstacles to the effective use of topical treatment (*Mounsey et al., 2016*). Oral therapy may eliminate some of the difficulties associated with topical treatment and potentially improve adherence, leading to better control of scabies in communities (*Mounsey et al., 2016*). However, we could not find any articles on the results of oral ivermectin treatment in the aftermath of earthquake disasters with poor hygiene and almost no access to water.

The aim of this study was to determine the impact of scabies after dermatologic complaints in people living in mass shelters such as tent cities and container cities after disasters and to investigate the efficacy of oral pharmacologic treatment on the disease when topical treatment could not be applied due to significant logistical concerns and treatment compliance problems.

## MATERIALS & METHODS

This cross-sectional study was conducted between 10 April 2023 and 10 May 2023 following receipt of ethical committee approval. This study was performed in line with the principles of the Declaration of Helsinki. Approval was granted by the Ethics Committee of Ankara Yıldırım Beyazıt University (2023/190). Verbal consent was obtained from the participants who participated in the study.

### Study areas

The research was carried out in eight mass accommodation areas in the central districts of Onikişubat and Dulkadiroglu in the province of Kahramanmaraş. According to Turkish Disaster and Emergency Management Authority (DEMA) records, the total population of the Kahramanmaraş central district tent cities and container cities is 43,183. These places are Kafum, Saim Cotur, university campus (six neighborhoods), 15 Temmuz Millet Bahcesi, Karaziyaret, Kuyumcukent, and Dogukent tent cities, and the Migrant Health Container City.

### Study design and sample size

Our study population consisted of 233 patients diagnosed with scabies by physicians between 10 April 2023 and 10 May 2023 and started on pharmacological (ivermectin) therapy recommended by the Ministry of Health among all the residents of the tent cities and container cities in this region. Data were collected from patients who applied to the health tents set up by the Ministry of Health in Kafum, Saim Cotur, the university campus (six neighborhoods), 15 Temmuz Millet Bahcesi, Karaziyaret, Kuyumcukent, and Dogukent tent cities, and the Migrant Health Container City. Our physicians have been assigned by the Ministry of Health and have received the necessary training to competently diagnose scabies after clinical examination. In uncertain cases, support was received from clinicians (or medical doctors) with extensive experience in diagnosing scabies. Individuals who were not diagnosed with scabies, declined to take part in the research, used treatments such as permethrin, malathion, benzyl benzoate, and sulfur ointment lotion, weighed <15 kg, or had active psychotic disorders, and pregnant women, were excluded from the study. No sample selection was employed, the aim being to contact the entire population.

Health tents were set up in all tent cities and container cities. Our health service was free and open 24 h a day. All citizens with medical complaints related to scabies were evaluated by our doctors without any obstacles. We visited the tents and containers where all the patients diagnosed with scabies were staying. Everyone staying under the same roof was examined for scabies. Again, our doctors working in each tent city or container city made daily planned home visits and provided mobile medical services to the entire population within a week. In addition to the exclusion criteria (21 weighing <15 kg, two pregnant women, and 5 using topical lotion), we completed our study with 192 patients after excluding individuals who did not want to be included in the study, discontinued the treatment (7), or could not comply with the treatment (6). A form consisting of 17 questions concerning sociodemographic characteristics, chronic skin disease (eczema, dermatitis, urticaria, *etc.*), symptoms related to existing scabies disease (pruritus, rash, a subcutaneous crawling sensation, fever, *etc.*), dates of initiation of first (day one) and second (day eight) dose pharmacological (ivermectin) therapy, severity of symptoms after treatment, and improvement in lesions was applied using the face-to-face interview method. Questions were scaled from 0 to 10 to determine the severity of symptoms of pruritus, rash, or a subcutaneous crawling sensation (with 0 being no symptoms and 10 being very severe).The dosing regimen was administered in two doses on the first and eighth days, according to the international standard treatment protocol. Doses were administered at 200 mcg/kg (*Mueller et al., 2019*). Undiagnosed but symptomatic individuals who shared the same environment as the cases received a single dose of oral pharmacological treatment. Additional hygiene recommendations for clothing, bedding, and linens were also given and implemented. The patient was interviewed face-to-face three times before treatment, on the fifth day after the first dose, and on the fifth day after the second dose, and symptoms and recovery status were investigated. For 19 illiterate people, our doctors read the questionnaire and responded according to the answers given.

## Statistical analysis

The study data were evaluated on SPSS software. Descriptive statistics were expressed as number (n), percentage (%), mean, median, and standard deviation (SD). Normality of distribution was evaluated using the Kolmogorov–Smirnov test. The chi-square test was applied in the comparison of categorical data, and one-way and two-way ANOVA in the analysis of numerical data. P-values <0.05 were considered significant.

## RESULTS

One hundred and ninety-two scabies patients (82.4%), 47.9% ($n = 92$) of whom were women, were included in the study after the exclusion criteria were applied. The frequency of scabies in the total population in tent cities and container cities was found to be at least 0.54%. The mean age of the study group was 24.75 ± 17.85 years (range 2–90). In terms of educational status, 29.7% ($n = 57$) of the cases had not received eight years of basic education, while 17.7% ($n = 34$) were university graduates or postgraduates. In addition, 82.8% ($n = 159$) of the patients lived in a tent city and 11.5% ($n = 22$) lived in a single tent

**Table 1** Patients' sociodemographic data, symptoms, and lesion results.

| | | n | Percentage (%) | | | | n | Percentage (%) |
|---|---|---|---|---|---|---|---|---|
| **Gender** | Female | 92 | 47.9 | | Fingers | Yes | 78 | 40.6 |
| | Male | 100 | 52.1 | | | No | 114 | 59.4 |
| **Education** | Illiterate | 19 | 9.9 | | Wrists | Yes | 96 | 50.0 |
| | Primary education | 38 | 19.8 | | | No | 96 | 50.0 |
| | Secondary education-High school | 101 | 52.6 | | Hands | Yes | 126 | 65.6 |
| | University or postgraduate | 34 | 17.7 | | | No | 66 | 34.4 |
| **Accommodation area** | Container | 4 | 2.1 | | Forearm | Yes | 144 | 75.0 |
| | Tent (single) | 18 | 9.4 | | | No | 48 | 25.0 |
| | Tent City | 159 | 82.8 | | Umbilicus | Yes | 167 | 87.0 |
| | Other | 11 | 5.7 | | | No | 25 | 13.0 |
| **Chronic skin disease** | Yes | 3 | 1.6 | **Lesion location** | Back | Yes | 135 | 70.3 |
| | No | 189 | 98.4 | | | No | 57 | 29.7 |
| **Treatment status(chronic skin disase)** | Yes | 3 | 1.6 | | Foot | Yes | 78 | 40.6 |
| | No | 189 | 98.4 | | | No | 114 | 59.4 |
| **Pruritis** | Yes | 190 | 99.0 | | Leg | Yes | 118 | 61.5 |
| | No | 2 | 1.0 | | | No | 74 | 38.5 |
| **Rash** | Yes | 188 | 97.9 | | Face | Yes | 30 | 15.6 |
| | No | 4 | 2.1 | | | No | 162 | 84.4 |
| **Subcutaneous crawling sensation** | Yes | 43 | 22.4 | | Neck | Yes | 23 | 12.0 |
| | No | 149 | 77.6 | | | No | 169 | 88.0 |
| **Fever** | Yes | 1 | 0.5 | | Scalp | Yes | 22 | 11.5 |
| | No | 191 | 99.5 | | | No | 170 | 88.5 |

or container. Three (1.6%) patients had chronic skin diseases and were receiving routine follow-up (Table 1).

The most common complaints were pruritus (99.0%) and rash (97.9%), and the least common was fever (0.5%). Lesion sites were most frequently in the umbilicus (87.0%), forearm (75.0%), and back (70.3%) and least commonly in the scalp (11.5%), neck (12.0%), and face (15.6%) (Table 1).

When the lesions were analyzed according to their morphology, papules were present in 80.70% of the study group ($n = 155$), eczematization in 82.3% ($n = 158$), excoriations in 21.9% ($n = 42$), vesicles in 15.6% ($n = 30$), and nodules in 15.10% ($n = 29$). The results are shown in Fig. 1.

The median scores (percentile 25–75) for pruritus and rash before treatment were 7.00 (6.00–8.00) and 6.00 (5.00–8.00), respectively.

The severity of pruritus after both the first and second doses of medication differed significantly compared to pre-treatment ($p < 0.001$). There was no significant difference in pruritis severity after the first treatment dose compared to after both doses of treatment ($p = 0.324$). The severity of rash after both the first and second doses of medication differed significantly compared to pre-treatment ($p < 0.001$). There was no significant difference in rash severity after the first treatment dose compared to after both doses of

**Table 2  Pairwise group comparisons of symptom severities before and after treatment.**

| | Samples 1–2 | Sample 1 severity | Sample 2 severity | Std. Test Statistic | P* |
|---|---|---|---|---|---|
| **Pruritus** | Pretreatment-after the first dose of pharmacological therapy | 6.65 ± 1.56 | 0.60 ± 1.51 | 13.625 | <0.001 |
| | Pretreatment-after the second dose of pharmacological therapy | 6.65 ± 1.56 | 0.22 ± 0.97 | 15.233 | <0.001 |
| | After the first dose of ivermectin-after the second dose of pharmacological therapy | 0.60 ± 1.51 | 0.22 ± 0.97 | 1.607 | 0.324 |
| **Rash** | Pretreatment-after the first dose of pharmacological therapy | 6.33 ± 1.66 | 0.60 ± 1.54 | 13.778 | <0.001 |
| | Pretreatment-after the second dose of pharmacological therapy | 6.33 ± 1.66 | 0.23 ± 1.03 | 15.386 | <0.001 |
| | After the first dose of ivermectin-after the second dose of pharmacological therapy | 0.60 ± 1.54 | 0.23 ± 1.03 | 1.607 | 0.324 |
| **Subcutaneous crawling sensation** | Pretreatment-after the first dose of pharmacological therapy | 1.26 ± 2.59 | 0.11 ± 0.86 | 2.705 | 0.021 |
| | Pretreatment-after the second dose of pharmacological therapy | 0.11 ± 0.86 | 0.04 ± 0.64 | 3.036 | 0.007 |
| | After the first dose of ivermectin-after the second dose of pharmacological therapy | 0.11 ± 0.86 | 0.04 ± 0.64 | 0.332 | 1.000 |

Notes.

*Two-way ANOVA.

treatment ($p = 0.324$). The severity of subcutaneous crawling sensations before treatment differed significantly from that after the first and second doses of treatment ($p = 0.021$ and $p = 0.007$, respectively). No significant difference was found between the severity of subcutaneous crawling sensations after the first and second doses of treatment ($p = 1.000$) (Table 2).

One hundred fifty-nine (82.8%) patients exhibited complete recovery after the first dose of pharmacological therapy, and 30 (15.6%) partial recovery. Three (1.6%) patients failed to improve. After the second dose of pharmacological treatment, 173 (90.1%) patients exhibited complete recovery, and 17 (8.9%) partial recovery. Two (1.0%) patients exhibited no improvement after two doses.

## DISCUSSION

Earthquakes are among the deadliest natural phenomena, directly affecting tens of thousands of lives. Following particularly devastating earthquakes, affected residents are transferred to communal accommodation areas by their governments to meet their shelter needs (*Al Mandhari, 2023*). The provision of basic needs such as heating, water, toilets, and showers in communal areas such as tents, tent cities, and container cities may be disrupted by damage to the urban infrastructure (*Villasana, 2023*). The emergence of scabies in such areas is facilitated by factors such as poor living conditions and poor hygiene (*Engelman & Steer, 2018*).

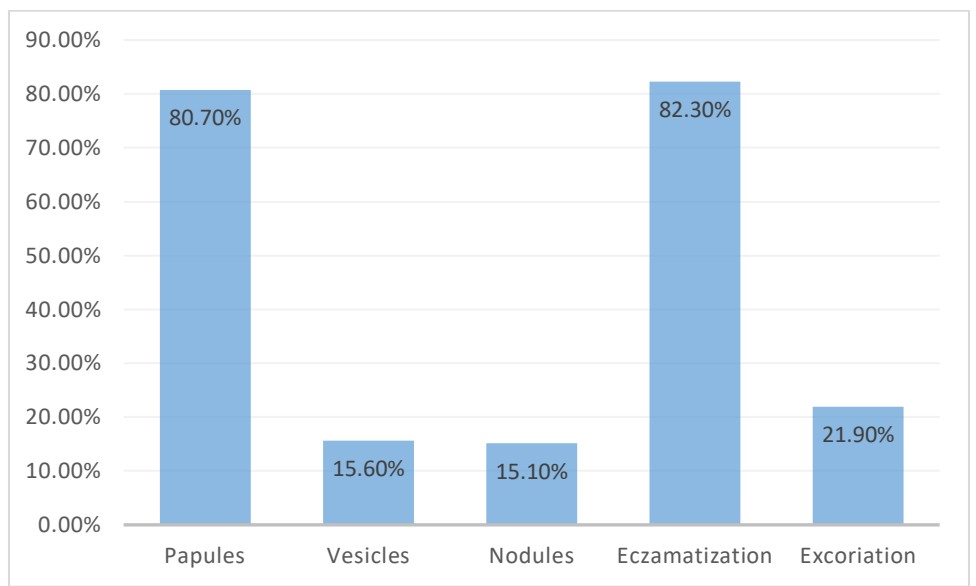

**Figure 1  Morphology of scabies lesions.**

In their study of disease patterns in communal living areas after a devastating earthquake in the northern part of Pakistan, Shah et al. reported that scabies exhibited the second highest common (17%) (*Shah et al., 2010*). In 2015, after the 7.8-magnitude earthquake in Nepal, the most common health complaints were insect bites (35.5%), followed by acute gastroenteritis (18.2%). Scabies was also common (16%) (*Malla et al., 2016*). A study conducted by *Yürekli et al. (2024)* showed a significant increase in the incidence of scabies after earthquakes. In the present study, scabies was detected in at least 5.4 out of every 1000 residents in communal accommodation areas during the specified periods. We could not make a clear comparison of the results of our study due to the different results in the literature and the rarity of post-earthquake scabies impact articles. More research needs to be done on this subject.

In terms of the disease pattern, the frequency of scabies in a study from Liberia was found to be higher in males than females. In that same study, *Collinson et al. (2020)* reported the lowest percentage of scabies (10%) in the group with the highest level of education. Callum and Karaca reported a higher frequency in men (*Callum et al., 2019*; *Karaca Ural, Çatak & Ağaoğlu, 2022*). Men constituted 42% ($n = 18,136$) of the tent cities and container cities included in our study. Despite this, more males (52%) were diagnosed with scabies in our study. Furthermore, when we look at the educational level of our patients, 17.7% were university or postgraduate level. This rate may be attributable to individuals with higher levels of education being more sensitive to hygiene, and possessing good socio-economic status, meaning that they are less commonly relocated to public accommodation areas.

The most common symptoms of scabies are pruritus and rash (*Jannic et al., 2018*; *Ständer & Ständer, 2021*). Leung's systematic review described papules and erythematous rashes as the most common morphologies (*Leung, Lam & Leong, 2020*). *Skayem et al. (2023)*

reported the hands, umbilicus, and forearm as the most commonly involved sites in their study from France. Pruritus (99.0%) and rash (97.9%) were the most common complaints among our patients. The most common sites in this study were the umbilicus (87.0%), forearm (75%), and back (70.3%). The most frequently detected lesion morphologies were papules (80.7%) and eczematization (82.3%).

A study on the treatment of scabies in Spain shows that although 5% permethrin has recently been accepted as first-line treatment, its efficacy against the disease is declining as resistance to the drug is increasing. However, oral ivermectin has been shown to be a first-line pharmaceutical therapy alternative (*Morgado-Carrasco, Piquero-Casals & Podlipnik, 2022*). *Lobo & Wheller (2021)* study on the roles of topical permethrin and ivermectin in cases of infantile scabies in Australia argued that oral ivermectin is a reliable modality and should even be preferred in cases resistant to permethrin. *Chiu & Argaez*'s *(2019)* study of the use of ivermectin in scabies in Canada suggested that oral ivermectin may be preferable in terms of clinical effectiveness and cost-effectiveness. In their 2019 algorithm for scabies outbreak management, *Mueller et al. (2019)* proposed two doses of oral ivermectin given one week apart. In another study conducted in Austria, it was reported that the efficacy of permethrin treatment decreased after the development of drug resistance, and oral ivermectin treatment may be preferred instead (*Meyersburg et al., 2023*). In the present study, the severity of pruritus and rash decreased significantly after oral pharmacological therapy compared to pretreatment levels ($p < 0.001$). No significant difference was observed in symptom severity after the first and second pharmacological therapy doses ($p = 0.324$, $p = 0.324$). *Dey, Agarwal & Sagar (2022)* found that topical placebo cream and oral ivermectin therapy exhibited 70% efficacy on the seventh day and 81% efficacy on the 14th day. The previous literature also suggests that the potency of oral ivermectin increases after dose repetition (*Behera et al., 2021*). Although the main effect on symptoms was achieved with a single dose of pharmacological therapy in the present study, complete recovery after two doses was observed in 99% of patients. After natural disasters, such as earthquakes, which can lead to poverty, mass shelters are built. It can be concluded that oral pharmacological treatment is an effective and reliable treatment method for scabies cases, especially in these periods due to poor hygiene conditions and difficulty accessing clean water.

## LIMITATIONS

We conducted our study in the post-disaster period, making it impossible to reach the entire population. It should be noted that 0.54% is the "estimated frequence" based on self-reporting of cases and follow-up of contacts, and that this value may be an underestimate due to the possibility that not all people with scabies infestation were identified during the study. In order to contribute to the literature, a control group can be added to our study, and comparisons can be made. Studies should be conducted on larger populations in the post-disaster period.

## CONCLUSION

In this study, the frequency of scabies was found to be at least 0.54% in the total population living in tent cities and container cities after the earthquake. It is likely that awareness, accurate diagnosis, and early initiation of oral pharmacological treatment in mass accommodation areas will in significantly reduce the spread of scabies and provide effective control. Even a single dose of oral pharmacological therapy resulted in an 82.8% ($n = 159$) complete cure rate. However, since ivermectin has no oocyte or larvicidal properties, we recommend the administration of two doses one week apart. The complete cure after two doses in this research was 99% ($n = 190$). In the light of the deterioration of urban infrastructure after major earthquakes, the irregular nature of mass accommodation areas, lack of hygiene, and the difficulty in accessing clean, usable water, oral pharmacological therapy may represent the first therapeutic option in terms of ease of use and effectiveness.

### Funding

The authors received no funding for this work.

### Competing Interests

The authors declare there are no competing interests.

### Author Contributions

- Muhammed Mustafa Beyoğlu conceived and designed the experiments, performed the experiments, analyzed the data, prepared figures and/or tables, authored or reviewed drafts of the article, and approved the final draft.
- Mehmet Enes Gokler conceived and designed the experiments, performed the experiments, analyzed the data, prepared figures and/or tables, and approved the final draft.

### Human Ethics

The following information was supplied relating to ethical approvals (i.e., approving body and any reference numbers):

Approval was granted by the Ethics Committee of Ankara Yıldırım Beyazıt University (Ethical Application Ref: 2023/190).

### Data Availability

The raw measurements are available in the Supplementary Files 1.

### Supplemental Information

Supplemental information for this article can be found online at http://dx.doi.org/10.7717/peerj.18242#supplemental-information.

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
