# Peer review of "The impact of scabies in tent cities in Kahramanmaraş after the Turkish earthquakes: oral pharmacologic treatment efficacy"

_PeerJ, doi:10.7717/peerj.18242_

## Round 0.1 · original submission · Major Revisions

· Academic Editor

Major Revisions

Dear Dr. Beyoğlu,

If you feel that you can accurately revise your paper in response to the reviewers' comments, please submit a revised version. Please also write a response to each of the reviewers' comments.

Yours,

Yoshi
Prof. Yoshinori Marunaka, M.D., Ph.D.


Reviewer 1 ·

Basic reporting

This manuscript reports a descriptive analysis of scabies prevalence and effectiveness of ivermectin in mass accommodation settings following a natural disaster. The manuscript is, for the most part, clearly written, and provides a useful additional data point on an important skin pathogen.

While the writing is generally clear, the introduction and discussion would benefit from some additional structure – there is some repetition between the two sections in terms of background information provided, some information (eg, on treatment) that is missing (or inadequate) in the introduction, and a need to more explicitly state the questions/discussion points that are being addressed. I have provided additional queries, comments, and suggestions on the manuscript below in the "Additional comments" section below.

Raw data have not been shared.

Experimental design

The description of the analysis is generally clear; however, there should be some additional comment on the limitations of the estimate of prevalence. As the study population only comprised patients treated for scabies, the prevalence of undiagnosed and untreated scabies in the broader population remains unknown.

I also had some questions about the application of the exclusion criteria and other aspects of reporting, detailed in the "Additional comments" section below.

Validity of the findings

Noting the point raised above about uncertainty of prevalence estimate, the presentation of the analysis otherwise seems appropriate.

However, I cannot see any indication of how the data could be obtained, therefore the results of the study cannot be replicated or verified. I note that it is an expectation of PeerJ that "All underlying data have been provided".

Additional comments

Please see below for specific comments on each section of the manuscript.

In addition, I have several additional questions that – if the data enables them to be addressed – would enhance the value of the study. I wonder if any further information on the environmental relationship between patients is available? At a minimum, can you report on variation in prevalence between the eight study sites? Do you know if patients were part of the same family unit, or were sharing the same tent / container?

Abstract

Line 16: Please add a sentence providing brief context for the study before stating the aim.

Line 25: “When evaluated with the tent city population” I wasn’t sure what the reason for this clause was – are you distinguishing “tent” from “container” here? Consider rephrasing.

Line 26 (and elsewhere): “the prevalence of scabies was 05.39%”. Is this correct? 233 cases in a population of 43,183 = 0.54%?


Introduction:

Line 54: what are the “times of transition” referred to? Is this the shift to temporary accommodation after a natural disaster? Please clarify.

Line 62: is the 300% increase in the Netherlands, in Europe, or globally? Please rephrase to clarify.

Line 63: “Earthquakes are one reason for this increase” – this needs more explanation: are earthquakes increasing in frequency or severity?

In general, the introduction contains most of the relevant introduction, I was left with a few questions:
- Dermatological diseases are mentioned as a particular problem following earthquakes and communal living. Is scabies the only such disease, or are there others?
- Oral pharmacological therapy is first mentioned in the final paragraph stating the aim of the study. It would be useful to briefly provide background on this in the preceding paragraph to help motivate the aim; that is, is it the standard therapy? Is it regularly used in this setting?


Materials & Methods:

Line 75 (and elsewhere): Please use names for months to avoid ambiguity (ie, I presume 10 April 2023, rather than 4 October 2023, but interpretation of day/month order varies).

Line 86: Why were patients not using permethrin lotion excluded from the study?

Line 90: Does the reference to “tent cities” imply that container city was excluded from the study?

Line 91: Was the study population reduced to 233 patients _after_ applying the exclusion criteria in lines 85-87? Or were these criteria used to reduce the number from 233 to 192 enrolled patients? Line 94 only mentions declining to participate as the reason for reduction.

Line 91: The fact that the study population comprised the 233 patients diagnosed with scabies (of ~43K total) suggests that it will be difficult to evaluate the frequency of scabies (as per aim in line 71). You have no way of estimating the number of people in the population who may have had a scabies infestation but not sought medical care, or who sought medical care but were not correctly diagnosed with scabies. Some estimate of these (eg, via a random sample of the population) would seem necessary to estimate the true prevalence of scabies in the total population.

Line 96: Can the survey be provided as supplementary information?


Results:

Line 114: What is the distinction between living in a tent city versus living in a single tent or container? Given that (as per the Background/Methods) the entire study population was drawn from a eight mass accommodation areas, what was the accommodation status of the remaining 11 participants?

Line 124: What are the mean scores referring to?

Line 128: Please use same sentence structure for rash as for pruritus (location of “differed significantly” is currently switched around, which is confusing to read). The sentences reporting subgroup analysis are confusing, as it sounds like you are talking about “after the first and second doses” (same as previous result) rather than “after the first dose only compared to after both doses” (which is what I _think_ you are referring to).


Discussion:

Much of the information provided here feels like additional background (ie, on prevalence of scabies in different settings) rather than discussion. Also, the values provided for comparison are somewhat inconsistent, some being from communal accommodation following natural disaster (eg, Pakistan & Nepal) with others from non-disaster settings (eg Netherlands).

I would suggest revising the discussion to make clear why the provided information has been included; that is, explicitly frame the discussion in terms of comparison of observed prevalence in this site with comparable (and more general) studies.

Line 162: Collinson 2020 reported higher prevalence in men in Liberia. You report that 52% of your study population was male; however, it isn’t possible to determine prevalence, as no information is provided on the ratio of men to women in the underlying population (ie, the 43K residents of the mass accommodation areas.

Line 176: I wonder again about the reason for excluding patients not using permethrin (line 86). It might be interesting to see if efficacy of oral ivermectin differed in groups who did / did not use permethrin.

Line 199: “is an effective and reliable alternative”: alternative to what? If to permethrin, this reinforces above query about why study explicitly focused on patients using both treatments.


Figures: I could not see a reference to either of the two figures in the manuscript. However, I’m not sure that either of them are necessary; the information is readily interpretable numerically.

Figure 1: As Yes + No = 100%, it is not necessary to show both bars. In fact, I think this information could be more clearly conveyed numerically in a table (with four entries)

Reviewer 2 ·

Basic reporting

Beyoglu et al, outline the effectiveness of ivermectin in the disaster setting after earthquakes in Turkey. While the sample size is relatively sized and gives good data that is of interest to medical practitioners, some aspects of the study design need increased clarification.

There seems little prior literature in this area. I think this should be highlighted as a key reason for conducting this research and I think it is an important area of study. Motives and the background of why this study was conducted need further clarification.

Some references in the discussion seem to lack relevance.

Discussion of limitations and what further research would be of use to add to this study is lacking and needs to be included.

English on the whole is reasonable but needs some minor editing.

References listed below are not reviewed and would have relevance (noting these might not have been available on drafting the article):

Tunalı V, Harman M, Özbilgin A. Investigation of Malaria, Leishmaniasis, and Scabies Risk after Earthquakes and Recommendations for Prevention. Turkiye Parazitol Derg. 2023 Dec 27;47(4):249-255. English. doi: 10.4274/tpd.galenos.2023.26122. PMID: 38149448.

Karaca Ural, Z., Çatak, B. & Ağaoğlu, E. Prevalence of Scabies in the Covid-19 Pandemic Period and Determination of Risk Factors for Scabies: a Hospital-Based Cross-Sectional Study in Northeast Turkey. Acta Parasit. 67, 802–808 (2022). https://doi.org/10.1007/s11686-022-00524-6

Yıldız Zeyrek F et al. (2023) Microbiology Australia 44(4), 197–201. doi:10.1071/MA23058

Experimental design

Needs to be described in more detail, some aspects of the design are not clear and some calculations are incorrect, which leads to issues in the subsequent discussion. The survey undertaking by participants, how this was administered and how clinical assessment was undertaken needs to be outlined better. Any loss to follow up is not detailed. Statements of prevalence can not really be made in a self-selecting population. As far as can be determined this is not a sample of a population with assessment of scabies but rather patients presenting to physicians (self-selection) and then being enrolled into the study if not meeting exclusion criteria.

Validity of the findings

There is no real control group or clear assessment of the baseline denominator. Ideally if a new treatment is being assessed for efficacy you need to compare this to the usual care: in this case for example application of topical therapy.

While the data is useful, discussion needs include more detail on limitations and what further the authors think will need to be done to support their question of how effective is oral therapy for scabies (compared to topical therapy), which I do not think is really fully addressed in this study. Further data like cost effectiveness and durability of effectiveness should also be commented on, or at least stated that these will be considerations in a “real-world” setting. The article does give a convicing argument of ivermectin being more effectrive then current therapies in this setting.

Overall, while I think this is useful data to publish and the authors have looked at an area of great need, I think the manuscript needs to be revised to give it better context and clarity.

Additional comments

General comments for review:

Lines 16-18: Which earthquakes, when/where were these tent cities? Useful to include in abstract. Has there been concern about using topical therapy in this situation?

Lines 24-34: Keep reporting to the same decimal point, e.g. for the forearm it should be 75.0% not 75%

Page 38-40: Not clear- example more how these numbers were obtained and what the percentages mean.

Line 50-51: “and food and water malnutrition”- suggest “malnutrition and water scarcity”

Line 63-64: “Earthquakes are one reason for this increase (Engelman, Steer, 2018)”- this could be expanded – why? I would not necessarily think this would lead to a sustained increased prevalence.

Page 56-63: A bit more context of this potential increased prevalence would be helpful. Is this because of increased migration from higher prevalence regions, specific refugee intakes or is scabies more generally increasing across society?

Line 36-73: Would be good to comment if there is any protocols/ standard practice of treating scabies in a disaster situation and whether there are any studies of this? In particular their effectiveness. From the introduction you outline the societal/ hygiene factors leading to potential increased scabies risk, but I do not get a clear impression of why you wanted to undertake this study? Was it because of prior experience of trying to use topical therapy with significant logistic concerns, was it because topical therapy has been shown to be ineffective? What is the standard guidelines based treatment recommendations in Turkey? Are there known other preventative measures that need to be considered? Also do you have data stating the baseline scabies prevalence (i.e. before the earthquakes) in the local area? If not state that this information is not available.

Line 80-87: This is not well written- please revise. Suggest state area of study- then separate sentence to discuss the 8 accommodation areas. Discuss the underlying demographics such as the population of this area and the camps here: not in the study design and sample size section.

Exclusion criteria should be in the study design section. It is not clear what you mean by – “not using treatments such as permethrin lotion”? Was anyone not using permethrin excluded?

Line 89-102: I take it the 192 was 233 minus patients that meet any exclusion criteria- can you elaborate on this more. i.e. how many were children <15kg, pregnant etc.
As above- to be clear, was ivermectin the routinely used treatment, was there any use of permethrin and where these excluded if using? It needs to be clearly outlined how the 233 number was obtained- where all physicians in the tent cities involved and how confident are you that they were reporting cases to you? Is scabies a notifiable condition to public health authorities? Was it all based on clinical diagnosis only?

Line 95-96: “A form consisting of 17 questions”- it might be useful to have this outlined – I understand that it is in the supplemetary material but a bit detail in the body of text would be useful. Also how was this practically carried out and were there any difficulties encounted? For the approx. 10% of illiterate patients, how was this form completed?

Line 98: “Dates of initiation of first and second dose pharmacological (ivermectin) therapy” How are these spaced? You do mention a week apart elsewhere but should be clearly stated. Any change in therapy for immunosuppressed/ heavily infected (crusted) scabies? How did you assess compliance and were people lost to follow up from dose 1 to the second dose?

Line 110: Make it clear that the 82.4% is the percentage of people were left for statistical analysis after exclusion criteria applied.

Line 111: 05.39% should be 5.39%

Line 113: “29.7% (n=7)”- how can n =7 ? Check as well- 17.7% (n=4)? I take it from later reading the 17.7% is a percentage of scabies diagnosed of the total population of those with a university qualification.

Line 118-120: Often scabies is found in the interweb spaces between fingers- any reason this was not commented on?

Line 121-123: How was this done? Was morphology assessed by individual physicians or were photos taken and a dermatologist or someone familiar to skin lesions assessed them all to help standardization? Please expand.

Line 124-125: Scores- what do these mean?

Line 126-127: How is this so if the ranges overlap?

Line 127-128: What subgroups?

Line 124-135: Please outline how the severity of each feature- pruritis, rash, crawling sensations were quantified to get a comparison for before and after- it is not clear to the reader! Outline any scoring or severity system used.

Line 139: “Complete and partial recovery”- I take it this was from the questionnaires or was this based on physician assessment

Line 151: One of the most…For context what was the highest prevalence disease and some of the other major health issues in this setting?

Line 153-154: “Van Deursen et al. determined a prevalence of scabies, which is rising in the Netherlands, of 2.6:1000 (Van Deursen et al., 2022).” Not sure why have you mentioned this- in what context? Is this the total population in the Netherlands – after an earthquake, which I do not think is common there? Local data would be more relevant if available, and if not maybe data from a comparable region/ setting. It seems a bit out of context to use this reference, so please outline if this was the best available information or look for a better reference.

Line 154-155: How was this number obtained? Be consistent in reporting: 5:1000 if keeping this in line with the prior sentence. Why do you state “approximately” rather than stating the actual number. It would be good to a unit-to-unit comparison of available other studies in a similar post natural disaster situation to get an idea of how significant the rate is in your study.

Line 156: “The high prevalence”- no real comparison here to get a good sense of how significant this prevalence is.

Line 161-163: Try and keep results in the result section not discussion section.

Line 163-166: Can you really say 17.7% is a low figure. This is the same figure as the university educated percentage in the sample. You can’t get a prevalence figure like this if you do not know the denominator!! Please review. Need to know: university educated AND scabies / total university educated population in the population of 43183. You also need to compare the percentage of each education group being housed in the different accommodation groups if you really want to advance these hypotheses.

Line 167: “According to studies in the literature”- redundant. Just state the most common symptoms are x and x (ref). Or “recent studies by x et al, symptom x and symptom y are the most prevalent in patients infected with scabies.

Line 174-175: Which literature?

Line 176-177: Reference this and state reasons why? Is it concern re increasing resistance or difficulty applying topical rx?

Line 184-185: So, if no difference in efficacy what are the advantages of ivermectin in the post disaster setting?

Line 186-187: Is this why you chose the dosing regimen in this study? Does that align with Turkish guidelines on how to dose ivermectin. Was it weight based dosing?

Line 188-189: “benzyl benzoate therapy with oral ivermectin may represent the treatment of choice- why?” More context please.

Line 196-198: The second dose is often used as an assurance against reinfection as well as treating any new emerging mites from larvae or egg forms. It should be noted whether other non-pharmacologic measures and treatment of all “household” or tent members was attempted despite difficulties with the circumstances. Scabies can often live on cloths / bedding / furniture and readily reinfect if this is not taken into account {noting that these methodology clarifications should be in the materials and methods section}

Line 200: Especially devastating earthquakes. Suggest change to: such as earthquakes, which can lead to poor..

Line 203-205: Without a comparator group, hard to definitely say this (likely can assume), although the individual effect of symptom resolution was very good.

Table1: check for errors. For example, fever is listed as 99.4%. Define what you considered as “skin disease”

Figure 1: as stated elsewhere – was there any standardization in how these where defined or was it as reported by individual physician assessment?

Figure 2: I do not think this adds much to the manuscript – consider deleting

---

## Round 0.2 · Minor Revisions

· Academic Editor

Minor Revisions

Dear Dr. Beyoğlu,

Please submit your manuscript revised according to the comments by reviewers.

Yours,

Yoshi

Prof. Yoshinori Marunaka, M.D., Ph.D.

Reviewer 1 ·

Basic reporting

The revisions have, in general, clarified the manuscript. However, there are a few new spelling/grammatical errors introduced. Please proofread carefully. (eg, in Study Design and Sample Size: "weighting" -> "weighing"; occurs multiple times).

Experimental design

The additional information and corrections have improved the manuscript. However, I still have a major concern about the use of the term "prevalence" to describe the observations made in the study.

If I understand correctly, all that you can conclude is that 0.54% of the tent/container city population presented to health services with scabies. You do not know how many people in the population may have experienced scabies but not presented to health services (and hence would not be included in your study sample).

Note that this does not detract from the study - the reports on the characteristics of this (self-selecting) sample are still relevant and of interest - however, the language used must be accurate.

Validity of the findings

Apart from as noted above re estimate of prevalence, the revisions appear to have addressed issues with incorrect figures.

I still see no reference to where or how the data may be obtained.

Reviewer 2 ·

Basic reporting

Beyoğlu have responded to most comments from my initial review. The paper reads better and is clearer. There is better context to the study.

Overall no major concerns, however I have attached some minor comments to attend to.

Experimental design

My main concern is the ongoing statement about prevalence, which I do not think as been adequtely addressed from the initial comments.

Were all 43183 in the study area assessed? This is hard to believe as it would have been logistically challenging. There is the statement "No sample selection was employed, the aim being to contact the entire population"- page 151. But no clarification if this was achieved. If for example only 30,000 people were assessed this would potentially change the estimate. I would still like clarity in the manuscript about this. If a small proportion of the population that was partly self selecting sample - such as people attending free medical clinics- then the authors need to state this and acknowlegment limitations. There is a possibility that with this study design true prevalence will not be obtained (as the selection process could be biased), and the number is likely to underpresent undignosed cases.

If all or close to the entire population was assessed then this is a major achievement and the study team has done very well, but please make this clear.

Otherwise I am happy with the descriptive statistics.

Validity of the findings

I think this is reasonable but I would focus the study more on the feasiblity of ivermectin and reasons why it will be useful in the post disaster setting rather then on prevalence - as per above comments.

Please see additional comments below.

Additional comments

Line 30-31: was 05.39%. “The prevalence of scabies in tent cities and container cities was found to be 0.54%” As previously stated in the first review – be careful of stating prevalence unless a true assessment of the whole population is undertaken that is not a bias sample. Please clarify methodology here. If this is the self-selected group that presented with scabies- likely it will underestimate true prevalence.
Line 68-69: “closely related to public health”. I think this should be “Scabies is an important dermatological disease of significant public health impact during these challenging times”
Line 71: Scabies is common skin diseases worldwide> is a common skin disease worldwide
Line 79-80: Greater impact if you can state this was resistance to topical therapies?
Page 83-84: Delete “from the Netherlands”
Page 85-87: “The increase in the number of collective living areas due to the frequency of earthquakes is one of the reasons for this increase (Engelman, Steer, 2018).”- I presume this ref does not relate to the Netherlands (not known to be an earthquake prone area), but this needs to be clearer! It reads straight after commenting on increased rates in the Netherlands.
Page 110-113: “The purpose of this study was to determine the prevalence of scabies after dermatologic complaints in people living in mass shelters such as tent cities and container cities after disasters.” I would suggest not using a technical term like prevalence here, given noted many limitations of determining this, unless methodology is clarified. Better to state “impact (or burden) of scabies with patients (or people) presenting with dermatologic complaints in tent/ container cities after disasters”
I wonder if the focus should be more in keeping of assessing the effectiveness of Ivermectin in the post disaster setting.
Page 147: “training on scabies”> training to competently diagnose scabies after clinical examination.
Page 147-148: “In cases of suspicion, support was received from competent academicians through active consultation”. I would reword. “academicians” is an unusual phase> In uncertain cases, support was received from clinicians (or medical doctors) with extensive experience in diagnosing scabies
Page 167-168: Suggest change: “Questions were scaled from 0 to 10 to determine the severity of symptoms of pruritus, rash, or a subcutaneous crawling sensation (with 0 being no symptoms and 10 being very severe).”
Page 183: At the start of the sentence this should probably be: P-values
Page 188-189: As per other comments- potentially not really a true prevalence. 0.54% is the known diagnosed percentage- there will likely be a bias to undercounting/ underrepresenting. Suggest considering rephasing.
Page 213-214: “According to subgroup comparisons, there was no significant difference in rash severity only in the first treatment dose compared to both doses of treatment (p=0.324).” Delete “According to subgroup comparisons”- repetitive. Suggest> “rash severity after the first treatment dose compared to after both doses of treatment”
Page 237-240: “In 2015, after the 7.8 magnitude earthquake in Nepal, the most common insect bite and acute gastroenteritis cases were seen in mass accommodation areas. In the same article, scabies was one of the most common infectious diseases (Malla et al., 2016).” Please rewrite. I take it you mean the most common health complaints were insect bites (stated prevalence / number), followed by X (%). Scabies was also common at..
Page 241-243: More useful if you actually state the findings rather than make a general statement- increased by how much etc. Do not need to state date if this is in the refence at the end of the sentence.
Page 246-247: As per previous statements.
Page 260: The 2019 is for one study, please delete. As per prior comment- I would suggest you do not state dates if this is clearly listed in the reference at the end of the sentence, it is repetitive.
Page 299-301: “Saborni et al. found that oral ivermectin therapy exhibited 70% efficacy on the seventh day and 81% efficacy on the 14th day (Dey, Agarwal & Sagar, 2022)” Please check this – the ref does not line up with the sentence.
Page 315-316: I’m not sure what you mean by adding a control group here. In an ideal world, to determine efficacy you would have a control group, but you have also stated the difficulties of not being able to do this given the circumstances.
Page 319: As per prior comments about prevalence. It is not disasters but this particular earthquake.
Page 322: “It can be assumed that awareness, accurate diagnosis, and early initiation of oral pharmacological treatment in mass accommodation areas will significantly reduce the spread of scabies and provide effective control”. I would probably say “it is likely that” rather then assume. Add will in before significantly.

---

## Round 0.3 · Minor Revisions

· Academic Editor

Minor Revisions

Please address these changes and resubmit. Although not a hard deadline please try to submit your revision within the proposed timeframe.

Yours,

Yoshi

Prof. Yoshinori Marunaka, M.D., Ph.D.

Reviewer 1 ·

Basic reporting

no comment

Experimental design

no comment

Validity of the findings

no comment

Additional comments

Thank you for considering reviewer comments. However, substituting "frequency" for "prevalence" does not, in my view, address the concern, as prevalence is simply a measure of frequency (see eg https://pubmed.ncbi.nlm.nih.gov/20173345/).

If I may make a more direct suggestion, I would you to clarify that that 0.54% is the "estimated prevalence" based on self-reporting of cases and follow-up of contacts (as per your newly added text) and to add a note to that this value may be an underestimate due to the possibility that not all people with scabies infestations were detected during the study.

This is a clear statement of how the figure should be interpreted, and does not detract from the message and relevance of the manuscript.

Reviewer 2 ·

Basic reporting

Mostly no issues. Please to check grammar - minor changes required.

Experimental design

As per previous explanative statements have been improved.

Validity of the findings

Improved compared to previous. I think interpretation is clearer.

Additional comments

Review response no 2.

I think the article is improved and happy with most changes. Just a few things to look at please.

Thanks for attempting to clarify the article, however substituting frequency for prevalence is not ideal as prevalence is a subset of disease frequency. i.e. I think you should use a less technical term such as “impact”. Frequency is a total count of an event over a certain time frame. Usually cumulative frequency can be represented as a percentage but this is essential taken at a point in time, as as per prior comments the denomenator is not clearly defined.

Suggest change to Impact in title and most of the text.

Consider changing “frequency” > impact or burden elsewhere if appropriate (although in some instances frequency will still suffice).

You also have substituted “frequency” for “prevalence” throughout the document and this is clearly not always appropriate!

Line 28-29 and 164-165 and 274-275: The frequency of scabies in tent cities and container cities was found to be 0.54%. > I would suggest stating some along the lines of: “This sample represented 0.54% of the total population of the Kahramanmara central district tent cities”.
Or “scabies affected at least 0.54% of the population…” as appropriate in the different sections depending on context.

Line 64: Is a common skin diseases worldwide> it is a common

Line 65-69: Use prevalence if this was used in the associated reference.

Line 72: As above.

Line 127: disorders, and were pregnant women, > why has the “were” been added? Take out.

Line 210: As per line 65-69 and 72.

---

## Round 0.4 · Minor Revisions

· Academic Editor

Minor Revisions

Two reviewers suggest very minor revision. I believe that the revision is very easy. Please submit your revised manuscript as soon as possible.

Yours,
Yoshi
Prof. Yoshinori Marunaka, M.D., Ph.D.

Reviewer 1 ·

Basic reporting

na

Experimental design

na

Validity of the findings

na

Additional comments

line 27: this sentence no longer makes sense. you need to explain what 0.54% refers to.

line 62: "It is a common affecting..." --> "It is a common skin disease affecting..."

line 159: same comment as above - you need to explain what 0.54% refers to.

line 201: this doesn't make sense; revert to "frequency" (you can't just do search and replace without checking your text!!!)

line 256: again, you need to explain what 0.54% refers to.

line 263: again.

Reviewer 2 ·

Basic reporting

Reads now much the same as prior, however I still have concerns as the authors have just replaced "frequency" with "impact", without taking into account the nuances and context of the statements! See below.

Experimental design

No change.

Validity of the findings

No change.

Additional comments

Line 27-28: You can’t just replace “frequency” with “impact” in every sentence. It depends on the context. Please revise. Like suggested: your findings indicate at least 0.54% of the population in in the tent cities were affected by scabies.
Line 57: As above! Along with 65, 70, 77….

---

## Round 0.5 · accepted · Accept

· Academic Editor

Accept

Congratulations again.
Yours,
Prof. Yoshinori Marunaka, M.D., Ph.D.